# Plasmid-Based Reverse Genetics System Enabling One-Step Generation of Genotype 3 Hepatitis E Virus

**DOI:** 10.3390/v17050669

**Published:** 2025-05-03

**Authors:** Tominari Kobayashi, Takashi Nishiyama, Kentaro Yamada, Kazumoto Murata, Hiroaki Okamoto

**Affiliations:** Division of Virology, Department of Infection and Immunity, Jichi Medical University School of Medicine, 3311-1 Yakushiji, Shimotsuke-Shi 329-0498, Tochigi, Japan; kobayashi-tm@jichi.ac.jp (T.K.); kentaro-y@cc.miyazaki-u.ac.jp (K.Y.); kmurata@jichi.ac.jp (K.M.)

**Keywords:** hepatitis E virus, plasmid-based reverse genetics system, mammalian expression promoter, ribozyme, vaccinia virus capping enzyme

## Abstract

Hepatitis E virus (HEV) is a positive-sense, single-stranded RNA virus that poses a significant public health risk, yet its study is hindered by the complexity of conventional RNA-based reverse genetics systems. These systems require multiple steps, including genome cloning, in vitro transcription, and capping, making them labor-intensive and susceptible to RNA degradation. In this study, we developed a single-step, plasmid-based HEV expression system that enabled direct intracellular transcription of the full-length HEV genome under a cytomegalovirus immediate-early (CMV-IE) promoter. The viral genome was flanked by hammerhead (HH) and hepatitis delta virus (HDV) ribozymes to ensure precise self-cleavage and the generation of authentic 5′ and 3′ termini. This system successfully supported HEV genome replication, viral protein expression, and progeny virion production at levels comparable to those obtained using in vitro-transcribed, capped HEV RNA. Additionally, a genetic marker introduced into the plasmid construct was stably retained in progeny virions, demonstrating the feasibility of targeted genetic modifications. However, plasmid-derived HEV exhibited delayed replication kinetics, likely due to the absence of an immediate 5′ cap. Attempts to enhance capping efficiency through co-expression of the vaccinia virus capping enzyme failed to improve HEV replication, suggesting that alternative strategies, such as optimizing the promoter design for capping, may be required. This plasmid-based HEV reverse genetics system simplifies the study of HEV replication and pathogenesis and provides a versatile platform for the genetic engineering of the HEV genome.

## 1. Introduction

Hepatitis E is an acute viral hepatitis caused by the hepatitis E virus (HEV) [1,2]. While generally self-limiting and rarely fatal, HEV infection exhibits a mortality rate of 0.5–4% in young adults, which increases to 20–30% in pregnant women [1,2]. In developing countries, HEV is primarily transmitted via contaminated water and the fecal–oral route [1,2], while in developed countries, transmission occurs through the consumption of raw or undercooked animal meat products, blood transfusion, and organ transplantation [3,4]. Additionally, immunocompromised individuals, including organ transplant recipients, patients with hematologic malignancies, and individuals infected with the human immunodeficiency virus (HIV), are at risk of developing a chronic HEV infection, which can result in cirrhosis or even liver failure [5,6]. HEV has been implicated in a variety of extrahepatic manifestations, including neurological and renal complications, adding to its clinical complexity [7].

HEV exists in two distinct particle forms: the membrane-unassociated non-enveloped form (neHEV), present in bile and feces, and the membrane-associated quasi-enveloped form (eHEV), which circulates in the bloodstream and is found in culture supernatants [8,9,10,11]. HEV is classified under the genus *Paslahepevirus* within the family *Hepeviridae* and subfamily *Orthohepevirinae* [12]. It is a small, spherical virus with a single-stranded, positive-sense RNA genome of approximately 7.2 kilobases (kb) in length [13]. The genome consists of a 5′-untranslated region (UTR) capped with 7-methylguanylate, three open reading frames (ORFs; ORF1, ORF2, and ORF3), and a 3′-UTR with a polyA tail [14,15]. ORF1 encodes a non-structural polyprotein essential for viral replication, which contains several functional domains, including a unique MetY domain that combines the two methyltransferase (Met) and Y domains as previously described, a fatty acid binding domain (FABD)-like domain, a hypervariable region, an X or macro domain, a helicase (Hel), and an RNA-dependent RNA polymerase (RdRp) [16,17,18] (Figure 1A). This polyprotein is translated from the full-length viral genome [13]. ORF2 encodes the capsid protein, while ORF3 encodes a multifunctional protein; both are translated from a bicistronic subgenomic RNA of approximately 2.2 kb [9]. The ORF2 protein exists in infectious, glycosylated, and cleaved forms. The glycosylated and cleaved forms are thought to serve as immunogenic decoys [19,20]. The ORF3 protein functions intracellularly, but it is also known to be present in eHEV particles [11]. Within the species *Paslahepevirus balayani,* eight genotypes of HEV have been identified, designated HEV-1 through HEV-8 [21]. Primarily, HEV-1 to HEV-4 are known to infect humans [22], with only a single documented case of HEV-7 infection reported in a human [23]. HEV-1 and HEV-2 are human-restricted genotypes, typically transmitted via fecally contaminated drinking water, and generally associated with self-limiting infections. In contrast, HEV-3 and HEV-4 are prevalent not only in industrialized countries but also in many developing countries due to their zoonotic feature and they have the potential to progress to chronic infection, particularly in immunocompromised individuals [24].

A specific class of RNA molecules, known as ribozymes, catalyze various biochemical reactions in a manner similar to protein enzymes [25]. One such reaction is the self-cleavage of RNA at specific sites [25,26]. The hammerhead (HH) ribozyme, a member of the small endonucleolytic ribozyme family, was originally discovered in subviral plant pathogens. Similarly, the hepatitis delta virus (HDV) ribozyme, another small endonucleolytic ribozyme, is derived from HDV, a satellite of the hepatitis B virus [26,27]. This class of ribozyme is widely distributed across eukaryotic genomes and has also been identified in several bacterial genomes [26]. These ribozymes are capable of cleaving their own phosphodiester backbone without requiring additional cofactors [26,27]. The HH and HDV ribozymes enable sequence-specific cleavage downstream and upstream of the ribozyme, respectively [26].

Plasmid-based reverse genetics systems have been developed for various viruses [28,29,30]. In most cases, these systems are established by introducing T7 RNA polymerase and constructs that express the viral genome under the control of a T7 promoter into host cells [30]. To enable the delivery of T7 RNA polymerase, genetically modified vaccinia viruses expressing T7 RNA polymerase [31,32], or cell lines stably expressing this polymerase, are frequently utilized [33]. Additionally, for viruses requiring a 5′ cap structure for genome functionality, capping enzymes derived from either the vaccinia virus [34,35] or the African swine fever virus [36] are co-introduced. Alternative reverse genetics approaches that do not rely on T7 RNA polymerase have also been reported [37,38,39]. These methods involve a single-step process in which viral genomes are expressed under the control of mammalian expression promoters, enabling direct virus production within host cells [37,38,39]. In these strategies, additional RNA sequences are incorporated at both termini of the expressed viral genome [37,38,39]. To address the issue of unwanted extraneous sequences at the 3′ end of the viral genome, a strategy employing the HDV ribozyme has been developed. This ribozyme enables autocatalytic cleavage at the 3′ end of the RNA transcript, thereby producing a precise viral genome terminus [38,40,41].

For HEV, most reported reverse genetics systems utilize in vitro-synthesized RNA [42]. To date, only one plasmid-based reverse genetics system has been reported [43]. This system employs T7 polymerase-expressing cells, a plasmid containing the HEV genome under the control of a T7 promoter, and a vaccinia virus-derived capping enzyme [43].

In this study, we aim to construct a plasmid-based reverse genetics system for HEV-3 by incorporating an HH ribozyme at the 5′ end and an HDV ribozyme at the 3′ end of the HEV genome. The full-length HEV genome, derived from the cell-culture-generated HEV-3 JE03-1760F strain [44] (the JE03-1760F_p13-3 strain [45]), is cloned into a mammalian expression plasmid under the control of a mammalian promoter. Transfection of this plasmid into host cells facilitates the transcription of the full-length HEV genome, leading to the production of the infectious virus over time.

## 2. Materials and Methods

### 2.1. Plasmids and Construction

In this study, we utilized the cell-culture-generated JE03-1760F_p13-3 strain (DDBJ/EMBL/GenBank accession number AB425830 [45]), which was derived from the JE03-1760F strain (accession number AB437316 [44]). The pJE03-1760F_p13-3 plasmid was constructed using the same methodology as pJE03-1760F/wt [46], as depicted in Figure 1B. Briefly, polymerase chain reaction (PCR)-amplified fragments from pJE03-1760F_p13-3 (designated as fragments 1–3; f1 to f3) and the polyAT7ϕ were digested with the indicated restriction enzymes (Figure 1B). The digested fragments were subsequently ligated and inserted into the HindII/NheI sites of a modified pUC19 vector (pUC19ΔAatISapI) [46] using T4 DNA ligase (TOYOBO, Osaka, Japan). To facilitate the ligation of fragments f2 and f3, an AflII restriction site (Figure 1B, asterisk) was introduced by replacing T with C at the nucleotide (nt) position 4784 and replacing G with T at nt 4786, ensuring no alteration in the encoded amino acid sequence, as previously described for pJE03-1760F/wt [46]. For the negative control, the pJE03-1760F_p13-3/GAA plasmid was constructed by replacing aspartic acid (Asp) residues at positions 1561 and 1562 with alanine in the conserved catalytic site of RdRp, achieved by substituting A with C at nt 4707 and at nt 4710, as described for pJE03-1760F/GAA [47] (Figure 1B).

The pHEK293 Ultra Expression Vector I was purchased from TaKaRa Bio (Shiga, Japan). This vector, used as the backbone for dHEV/wt, contains a transactivation response (TAR) element and enables high-level recombinant protein expression via a transcriptional activation mechanism derived from the HIV-1 TAR-Tat system, under the control of a cytomegalovirus immediate-early (CMV-IE) promoter. The TAR element is an RNA sequence originating from HIV-1 that forms a stem-loop structure at the 5′ end of the viral RNA. The Tat (transactivator of transcription) protein is an RNA-binding transactivator that specifically interacts with the TAR element to enhance transcriptional activity. The HH ribozyme (Figure 1C) was synthesized as an oligonucleotide DNA (PCR primer, see below), while the HDV ribozyme (Figure 1C) was generated by annealing a pair of complementary oligonucleotides. Expression plasmids for the vaccinia virus capping enzyme, pCAG-D1R and pCAG-D12L, were obtained from Addgene (plasmid #89160 and #89161, respectively; Watertown, MA, USA) [34].

The full-length cDNA of the pJE03-1760F_p13-3 strain, incorporating the HH ribozyme at the 5′ end and the HDV ribozyme at the 3′ end, was cloned into the pHEK293 plasmid and designated as the dHEV/wt plasmid (Figure 1C). The construction process was as follows: First, the pHEK293 plasmid was modified to introduce additional restriction enzyme sites (BamHI-SpeI-NheI-XbaI-SalI) into the multiple cloning site. This was achieved by digesting the plasmid with BamHI/SalI and ligating it with an annealed oligonucleotide pair containing the restriction sites (5′-GATCCACTAGTGCTAGCACGCGTTCTACAG-3′ [sense] and 5′-TCGACTCTAGAGCTAGCACGCGTACTAGTG-3′ [antisense]) using T4 DNA ligase. Next, the HDV ribozyme [48] was inserted into the modified pHEK293 plasmid by digesting it with NheI/XbaI and ligating it with an HDV ribozyme fragment containing NheI/XbaI sites at its 5′- and 3′-terminal, respectively. The HDV ribozyme fragment was generated by annealing the oligonucleotide pair (5′-CTAGCGGCCGGCATGGTCCCAGCCTCCTCGCTGGCGCCGGCTGGGCAACATTCCGAGGGGACCGTCCCCTCGGTAATGGCGAATGGGACCCAT-3′ [sense] and 5′-CTAGA TGGGTCCCATTCGCCATTACCGAGGGGACGGTCCCCTCGGAATGTTGCCCAGCCGGCGCCAGCGAGGAGGCTGGGACCATGCCGGCCG-3′ [antisense]) and ligating it with T4 DNA ligase.

Subsequently, the 3′-terminal HEV fragment (698–7226 nt) with a 31 nt polyA tail and T7 terminator was excised from the pJE03-1760F_p13-3 using SpeI/NheI and inserted into the SpeI/NheI-digested HDV ribozyme-containing pHEK293 plasmid. The BamHI-HH ribozyme-5′-HEV-SpeI fragment, containing the 5′-terminal HEV genome fragment (nt 1–698) and the HH ribozyme [49] at the 5′ end, was PCR-amplified using pJE03-1760F_p13-3 as the template and a specific primer pair (5′-TTTTGGATCC ATCGACCACATGCGTGGTCTGCCTGATGAGGCCGAAAGGCCGAAAACCCGGTATCCCGGGTTCGCAGACCACGCATGTGGTCG-3′ [forward, with the HEV sequence corresponding to nt 1–20 underlined] and 5′-CCGGCACTAGTATCACCCTC-3′ [reverse, corresponding to nt 689–708]). The amplified fragment was digested with BamHI/SpeI and then inserted into the BamHI/SpeI-digested pHEK293 plasmid containing the 3′-HEV and HDV ribozyme sequences. The resulting plasmid (dHEV/wt-T7ϕ) contained a T7 terminator derived from pJE03-1760F_p13_3 immediately downstream of the polyA sequence.

To eliminate the T7 terminator and directly link the polyA sequence with the HDV ribozyme, the following steps were undertaken: The HEV-HDV ribozyme fragment was amplified using the pJE03-1760F_p13_3 plasmid as the template and a specific primer pair (5′-GGCTGGTTATCCTTATAACTACAATACAAC-3′ [forward, corresponding to nt 6833–6862] and 5′-GCAGGTCGACTCTAGATGGGTCCCATTCGCCATTACCGAGGGGACGGTCCCCTCGGAATGTTGCCCAGCCGGCGCCAGCGAGGAGGCTGGGACCATGCCGGCCTTTTTTTTTTTTTTTTTTTTTTTTTTTTTTTCCAGGGAGCGCGAAAAGC-3′ [reverse, the HEV sequence with a 31 nt polyA tail of nt 7209–7257 underlined]). The dHEV/wt-T7ϕ plasmid was digested with PsiI and XbaI, and the resulting fragments were fused using an In-Fusion HD cloning kit (TaKaRa Bio), according to the manufacturer’s protocol. As a genetic marker, a NaeI restriction site was introduced by replacing A with C at nt 706 and T with C at nt 709, without affecting the amino acid sequence. The PCR fragment containing the NaeI site was amplified using pJE03-1760F_p13_3 plasmid as the template and a specific primer pair (5′-TGATACTAGTGCcGGcTACAACC-3′ [forward, corresponding to nt 694–715; the SpeI recognition sequence is underlined, and mutated nucleotides are indicated in lowercase] and 5′-GGGTTCTGCTCAACCCTATAGTC-3′ [reverse, corresponding to nt 2624–2646]). The amplified fragment was then substituted into the original plasmid using the SpeI/NotI sites. In this manner, the final dHEV/wt plasmid was obtained. For the negative control, the dHEV/GAA plasmid was constructed by replacing aspartic acid (Asp) residues at positions 1561 and 1562 with alanine in the conserved catalytic site of RdRp, achieved by substituting A with C at nt 4707 and at nt 4710 (Figure 1D). Additionally, the construct lacking the HDV ribozyme (dHEV/ΔHDVrbz) was generated by excising the HDV ribozyme from the dHEV/wt plasmid using NheI and XbaI, followed by self-ligation using T4 DNA ligase (Figure 1D). All restriction enzymes were purchased from New England Biolabs (Ipswich, MA, USA). Sequencing was performed using the Applied Biosystems 3130xl Genetic Analyzer (Thermo Fisher Scientific, Waltham, MA, USA) in conjunction with the BigDye Terminator v3.1 Cycle Sequencing Kit (Thermo Fisher Scientific) or by Eurofins Genomics (Tokyo, Japan). A sequence analysis was performed using the Genetyx software program (version 13; Nihon Server, Tokyo, Japan) or A Plasmid Editor (ApE) software (version 3.1.4) [50].

### 2.2. In Vitro RNA Synthesis and Capping

Plasmid DNA (pJE03-1760F_p13-3) was linearized with the restriction enzyme BamHI-HF (New England Biolabs). In vitro transcription of UC-rHEV RNA was performed from the linearized pJE03-1760F_p13-3 plasmid using the AmpriScribe™ T7-*Flash*™ Transcription Kit (Epicentre/Illumina, Inc., San Diego, CA, USA), following the manufacturer’s protocol. The synthesized UC-rHEV RNA was purified via standard phenol–chloroform extraction and ethanol precipitation. To generate capped RNA (C-rHEV), the purified UC-rHEV RNA was subjected to enzymatic capping using the ScriptCap™ m^7^G Capping System (CELLSCRIPT, Madison, WI, USA) according to the manufacturer’s instructions. The capped RNA was subsequently purified by phenol–chloroform extraction and ethanol precipitation.

### 2.3. Cell Culture and Transfection

PLC/PRF/5 cells (ATCC CRL-8024; American Type Culture Collection, Manassas, VA, USA) were grown in Dulbecco’s modified Eagle’s medium (DMEM; 12800-058, Gibco/Thermo Fisher Scientific, Waltham, MA, USA) supplemented with 10% (*v*/*v*) heat-inactivated fetal bovine serum (FBS; 10270, Gibco/Thermo Fisher Scientific., Waltham, MA, USA), 100 U/mL penicillin, 100 µg/mL streptomycin, and 2.5 µg/mL amphotericin B (growth medium). The cells were maintained at 37 °C in a humidified atmosphere with 5% CO_2_.

For the transfection experiments, PLC/PRF/5 cells (1 × 10^4^) were seeded into 24-well plates (BioLite 24 Well Multidish, 130186; Thermo Fisher Scientific) one day before transfection. RNA transfection was carried out using 500 ng of RNA and the *Trans*IT-mRNA Transfection Kit (Mirus Bio LLC., Madison, WI, USA) according to the manufacturer’s instructions. For plasmid DNA transfection, 500 ng of plasmid DNA were transfected using one of the following transfection reagents: *Trans*IT-LT1 (Mirus Bio), Lipofectamine 3000 (Thermo Fisher Scientific), FuGENE HD (Promega, Madison, WI, USA), X-tremeGENE HP DNA (Roche, Basel, Switzerland), Xfect (Clontech, Takara, Mountain View, California, USA), polyethylenimine (PEI Max; Polysciences, Warrington, PA, USA), or EcoTransfect (OZ Bioscience, France), following the respective manufacturer’s protocols.

For the co-transfection of dHEV/wt or dHEV/GAA plasmids with vaccinia virus capping enzyme expression plasmids (pCAG-D1R and pCAG-D12L), 250 ng of dHEV/wt or dHEV/GAA plasmid DNA, along with 125 ng each of the pCAG-D1R and pCAG-D12L plasmids, were transfected into PLC/PRF/5 cells using *Trans*IT-LT1 reagent. Two days post-transfection, cells were washed five times with phosphate-buffered saline (PBS), and 500 μL of fresh growth medium was added. Thereafter, half of the culture medium was replaced with fresh medium every other day. Collected culture supernatants were centrifuged at 1300× *g* for 2 min at 4 °C, and the resulting supernatants were stored at −80 °C for the following experiments.

### 2.4. Quantification of HEV RNA

Viral genomic HEV RNA associated with viral particles were purified from the collected culture supernatants using TRIzol-LS Reagent (Thermo Fisher Scientific) according to the manufacturer’s protocols. HEV RNA was quantified using reverse transcription (RT)–quantitative polymerase chain reaction (qPCR) with the QuantiTect Probe RT-PCR Kit (Qiagen, Hilden, Germany). The assay utilized a specific primer set and a TaqMan probe targeting the overlapping region of ORF2 and ORF3, as previously described [8]. RT-qPCR reactions were performed on either the 7900HT Fast Real-Time PCR System (Applied Biosystems, Foster City, CA, USA) or the LightCycler 96 system (Roche Diagnostics KK, Tokyo, Japan).

### 2.5. The 5′ Rapid Amplification of cDNA Ends (5′RACE)

Culture supernatants from cells transfected with either RNA or plasmid DNA were treated with RNase or DNaseI, respectively. Particle-associated RNA was then purified from the treated supernatants using TRIzol-LS Reagent, following the manufacturer’s instructions. The purified RNA was reverse transcribed into cDNA using SuperScript II reverse transcriptase (Thermo Fisher Scientifics) and an HEV-specific antisense primer (5′-GGCCGAACCACCACAGCATTCG-3′; nt 111–132), according to the manufacturer’s protocol. The resulting cDNA was purified by standard phenol–chloroform extraction followed by ethanol precipitation. Subsequently, the purified cDNA was polyG-tagged via a terminal deoxynucleotidyl transferase (TdT) reaction (New England Biolabs). PolyG-tagged cDNA was amplified through two rounds of PCR with specific primer pairs. The first round of PCR was performed with the forward primer 5′-AAGGATCCGTCGACATCGATAATACGCCCCCCCCCCCCCCC-3′ (non-specific sequence underlined) and the reverse primer 5′-CCACCACAGCATTCGCCAAG-3′ (nt 106–125). The second round of PCR utilized the forward primer 5′-AAGGATCCGTCGACATCGAT-3′ (identical to the non-specific sequence of the first-round forward primer) and the reverse primer 5′- CTCAATGGCAGTAGTAATGCCAG-3′ (nt 54–76). The amplified DNA fragments were subsequently purified and subjected to sequencing.

### 2.6. Detection of Genetic Marker

Particle-associated RNA was purified from the culture supernatant of plasmid DNA- or RNA-transfected cells using TRIzol-LS Reagent, following the manufacturer’s protocol. Reverse transfection was performed using SuperScript II reverse transcriptase and an HEV-specific antisense primer (5′-CTGGCGGCCGGGGATGTAGTCACG-3′, nt 1283–1306), according to the manufacturer’s instructions. A specific 881 base pair (bp) region (nt 416–1296) was amplified from the cDNA using PCR with the primer pair: 5′-CGTCGTTCTGCTCTACGTGG-3′ (forward, nt 416–435) and 5′-GGGATGTAGTCACGGCCAGACTTCTC-3′ (reverse, nt 1271–1296). The resulting PCR products were digested with the NaeI restriction enzyme (New England Biolabs). Digestion produced fragments of 291 bp and 590 bp for plasmid DNA-derived HEV, whereas RNA-derived HEV remained undigested (881 bp).

### 2.7. Western Blotting

The culture supernatant from RNA- or plasmid-transfected cells was collected at 20 days post-transfection (dpt) and was subsequently boiled at 95 °C for 5 min in SDS sample buffer (final concentration: 60 mM Tris-HCl, 2% sodium dodecyl sulfate [SDS], 5% glycerol, 0.2% bromophenol blue [BPB], 5% 2-mercaptoethanol [2-ME], and 100 mM dithiothreitol [DTT]). The prepared samples were separated on a 12.5% polyacrylamide gel and then transferred onto Immobilon-P membranes (Millipore Corporation, Billerica, MA, USA) by electroblotting. The membrane was blocked with PBS containing 0.1% (*v*/*v*) Tween-20 (PBS-T) and 5% skim milk (BD Sciences, San Jose, CA, USA) and incubated with primary antibodies (anti-ORF2: H6210 [8], 1 μg/mL; anti-ORF3: TA0536 [51], 1 μg/mL) at room temperature for 1 h. Following washing with PBS-T, the membranes were incubated with an HRP-conjugated secondary antibody (GE Healthcare Japan, Tokyo, Japan; 1:20,000) at room temperature for 1 h. Chemiluminescent signals were developed using ECL solution and were detected with an ImageQuant LAS500 system (GE Healthcare Japan). Precision Plus Protein Dual Color Standards (Bio-Rad Laboratories, Tokyo, Japan) were used as molecular weight markers.

### 2.8. Immunofluorescence

RNA- or plasmid DNA-transfected cells (24 dpt) were re-seeded onto 8-well chamber slides (Thermo Fisher Scientific), washed with PBS, and fixed with 4% (*v*/*v*) paraformaldehyde in PBS at room temperature for 20 min. The cells were then permeabilized with 0.2% (*v*/*v*) Triton X-100 at room temperature for 10 min. After washing, the cells were incubated with primary antibodies (anti-ORF2: H6210 [8], 1 μg/mL; anti-ORF3: TA0536 [51], 1 μg/mL) at 37 °C for 1 h, followed by incubation with an Alexa Fluor 488-conjugated secondary antibody (Molecular Probes, Thermo Fisher Scientific; 1:2000) at 37 °C for 1 h. Nuclei were counterstained with 4′,6-diamidino-2-phenylindole (DAPI; Roche Diagnostics). Stained cells were visualized using an FV1000 confocal laser microscope (Olympus, Tokyo, Japan).

### 2.9. Northern Blotting

Total RNA was extracted from cells transfected with RNA- or plasmid DNA-transfected cells (24 dpt). RNA samples (1 μg) were separated on a 1% agarose gel and transferred onto a Hybond-N nylon membrane (GE Healthcare Japan). Following pre-hybridization, the membrane was hybridized overnight at 68 °C under high-stringency conditions using a DIG-UTP-labeled RNA antisense or sense probe specific to the HEV ORF2 sequence of the JE03-1760F genome (nt 6700-7100 or 5770-6520, respectively) [44], complementary to antigenomic or genomic RNA. Post-hybridization washes were performed stepwise with buffers containing 0.1–2× SSPE, 3M sodium chloride, 200 mM sodium phosphate, 20 mM EDTA, and/or 0.1% SDS. The target RNA was detected using a DIG Luminescent Detection Kit (Roche Applied Science, Penzberg, Germany) and was visualized with an ImageQuant LAS500 (GE Healthcare Japan). DynaMarker™ Prestain Marker for RNA High (BioDynamics Laboratory, Inc., Tokyo, Japan) was used as a molecular weight marker.

### 2.10. Virus Inoculation

The human lung carcinoma cell line A549 (No. RCB0098; RIKEN BRC Cell Bank, Tsukuba, Japan), which supports HEV propagation [44], was cultured in growth medium. Upon reaching confluence, the cells were trypsinized, diluted at a ratio of 1:4 in fresh medium, and seeded at 2 mL per well in a 6-well microplate (IWAKI, Tsukuba, Japan) one day prior to infection [45]. On the day of infection, A549 monolayers were washed three times with 1 mL of PBS without Ca^2+^ and Mg^2+^ [PBS(-)]. A 0.2 mL aliquot of culture supernatant, filtered through a 0.22 μm membrane, was then inoculated onto the monolayers (1 × 10^5^ copies/well). After incubation at room temperature for 1 h, the inoculum was removed, and 2 mL of maintenance medium was added. This medium comprised a 1:1 mixture of DMED and Medium 199 (Invitrogen) supplemented with 2% (*v*/*v*) FBS and 30 mM MgCl_2_, with all other components identical to the growth medium. Cultures were maintained at 35.5 °C in a humidified 5% CO_2_ atmosphere. The following day, infected cells were washed five times with 1 mL of PBS(-), and 2 mL of fresh maintenance medium was added. From day 2 post-inoculation, 1 mL of the culture medium was replaced with fresh maintenance medium every other day. Collected culture supernatants were centrifuged at 1300× *g* for 2 min at 4 °C, and the resulting supernatants were stored at −80 °C for subsequent experiments.

## 3. Results

### 3.1. Construction of a Plasmid-Based HEV Expression Vector

To develop a plasmid-based system for HEV expression, we first constructed the pJE03-1760F_p13-3 plasmid, which harbored the full-length genome of the cell-culture-adapted HEV-3 JE03-1760F strain [45]. This plasmid was generated using the same methodology as the previously established infectious cDNA clone of the wild-type JE03-1760F strain (pJE03-1760F/wt) [46], as illustrated in Figure 1B.

To further establish a plasmid-based reverse genetics system for HEV, we constructed the dHEV/wt plasmid (Figure 1C) using pJE03-1760F_p13-3 as the template. The pHEK293 vector was first modified to include additional restriction sites, after which the HDV ribozyme was inserted via NheI/XbaI cloning. The 3′-terminal HEV genomic fragment (nt 698–7226), containing a 31 nt polyA tail and a T7 terminator, was excised from pJE03-1760F_p13-3 and ligated into the modified vector. Subsequently, the 5′-terminal HEV fragment (nt 1–698), fused to an HH ribozyme, was PCR-amplified and inserted upstream of the 3′ fragment. To eliminate the T7 terminator, a replacement fragment was introduced by In-Fusion cloning, enabling a direct linkage between the polyA sequence and the HDV ribozyme. Silent mutations introducing a NaeI restriction site were incorporated at nt 706 and 709 as genetic markers. The assembled dHEV/wt plasmid was verified by sequencing.

For use as a negative control, we constructed the dHEV/GAA plasmid by replacing aspartic acid residues at positions 1561 and 1562 with alanine within the conserved catalytic domain of RdRp in ORF1 (Figure 1D). Furthermore, to generate a construct lacking the HDV ribozyme (dHEV/ΔHDVrbz), the HDV ribozyme sequence was excised from dHEV/wt using NheI and XbaI digestion, followed by self-ligation (Figure 1D). Details of these procedures are provided in Section 2.

### 3.2. Evaluation of Transfection Reagents

To determine the optimal transfection reagent for introducing the plasmid-based HEV expression vector into cells, a comparative evaluation was conducted. The dHEV/wt plasmid was transfected into PLC/PRF/5 cells using various transfection reagents, including *Trans*IT-LT1, Lipofectamine 3000, FuGENE HD, X-treamGENE, Xfect, Polyethylenimine (PEI), and EcoTransfect. The amount of HEV RNA released in the culture supernatant was measured over time. The efficiency of HEV production varied significantly among the different transfection reagents, with *Trans*IT-LT1 demonstrating the highest efficiency (Figure 2A). Consequently, *Trans*IT-LT1 was selected as the transfection reagent for all subsequent plasmid transfection experiments.

### 3.3. Comparison of HEV Production Efficiency Between the HEV Expression Plasmid Vector and In Vitro-Synthesized HEV RNA Genome

To compare the efficiency of HEV production between dHEV/wt and in vitro-synthesized HEV RNA genomes with or without a 5′ cap, PLC/PRF/5 cells were transfected with dHEV/wt, capped HEV RNA genome (C-rHEV), and uncapped HEV RNA genome (UC-rHEV). As a negative control for dHEV/wt, a mutant construct (dHEV/GAA) in which the GDD catalytic motif of RdRp was substituted with GAA was used. As a negative control for C-rHEV and UC-rHEV, a mutant construct (C-rHEV/GAA) with a GAA mutation was also used. The results indicated that both dHEV/wt and UC-rHEV exhibited a delayed increase in HEV RNA levels compared to C-rHEV, although their overall expression patterns were similar. Despite these temporal differences, the peak HEV RNA levels of all three constructs were comparable (Figure 2B). In contrast, no HEV replication was observed in cells transfected with dHEV/GAA and C-rHEV/GAA (Figure 2B).

Additionally, the impact of the HDV ribozyme located downstream of the polyA signal on HEV genome amplification was examined using a dHEV/wt variant lacking the HDV ribozyme (dHEV/ΔHDVrbz). Similarly to dHEV/GAA, dHEV/ΔHDVrbz exhibited markedly reduced levels of HEV RNA expression. These findings suggest that the presence of extraneous plasmid-derived sequences downstream of the poly A tail at the 3′ end of the HEV genome inhibits the intracellular amplification of the HEV genome.

### 3.4. Delayed Capping of Plasmid-Derived HEV Genomic RNA in Cells

To examine the temporal dynamics of capping in HEV particle-associated RNA genomes over time, we analyzed the HEV RNA genomes present in viral particles within the culture supernatant collected at 4, 8, and 16 dpt using 5′ RACE followed by sequencing (Figure 3). At 4 dpt, the HEV genome within the viral particles from the culture supernatant of cells transfected with a capped HEV RNA genome (C-rHEV) retained its cap structure, whereas genomes from dHEV/wt and UC-rHEV remained uncapped. By 8 dpt, the HEV genomes associated with viral particles from all three constructs had undergone capping. These findings suggest that the HEV genome transcribed from dHEV/wt within transfected cells was initially uncapped and, similar to UC-rHEV, acquired a cap structure as genome replication progressed.

### 3.5. Retention of the Genetic Marker in Plasmid-Derived HEV Genome

The HEV genome incorporated into the plasmid-based HEV expression vector contained a NaeI restriction enzyme recognition site introduced via a nucleotide substitution (Figure 1C). To assess the retention of this genetic marker, HEV particle-associated RNA was purified from the culture supernatant of dHEV/wt- and C-rHEV-transfected cells at 20 dpt, reverse transcribed, and subjected to restriction enzyme analysis. A PCR was performed to amplify an 881 bp region encompassing the NaeI site, followed by digestion with the NaeI restriction enzyme. The HEV genome derived from C-rHEV remained intact following NaeI treatment, whereas the genome derived from dHEV/wt was completely digested (Figure 4). These results demonstrated that the replicated HEV genome retained the original sequence derived from the plasmid-based HEV expression construct.

### 3.6. Expression of Viral Proteins from a Plasmid-Based HEV Expression Vector

To determine whether viral proteins were expressed in cells transfected with dHEV/wt, we assessed the presence of ORF2 and ORF3 proteins in the culture supernatant collected at 20 dpt using Western blot analysis. The expression of ORF2 and ORF3 was detected in culture supernatants from dHEV/wt-, C-rHEV-, and UC-rHEV-transfected cells. In contrast, however, no protein expression was observed in the negative control (mock) or culture supernatants from cells transfected with dHEV/GAA or dHEV/ΔHDVrbz, the latter lacking the HDV ribozyme at the 3′ end (Figure 5A). Similarly, immunofluorescence analysis of cells at 24 dpt revealed intracellular ORF2 and ORF3 protein expression in cells transfected with dHEV/wt, C-rHEV, and UC-rHEV, but not in dHEV/GAA- or dHEV/ΔHDVrbz-transfected cells (Figure 5B). These findings confirmed that viral proteins were efficiently expressed from the plasmid-based HEV expression system, contingent on the presence of functional RNA elements.

### 3.7. Replication of the Cellular HEV RNA Genome in HEV Expression Plasmid DNA-Transfected Cells

To verify the replication of the HEV genome in HEV expression plasmid DNA-transfected cells (24 dpt), cellular HEV genomes were analyzed by Northern blotting (Figure 5C). When hybridized with an antisense probe (Upper, Plus strand), both full-length HEV and subgenomic RNA genomes were detected in dHEV/wt, C-rHEV, and UC-rHEV at comparable levels. Similarly, a sense probe (Lower, Minus strand) revealed the presence of double-stranded RNA (dsRNA), indicative of the replication complex, in dHEV/wt, C-rHEV, and UC-rHEV (Figure 5C. These findings suggested that the HEV genome derived from plasmid-based HEV replicated in a manner similar to that of in vitro-transcribed HEV RNA.

### 3.8. Infectivity of Progeny Virions from Plasmid-Based HEV

To evaluate the infectivity of progeny virions, supernatants containing HEV particles derived from dHEV/wt (20 dpt) were inoculated into A549 cells. The replication kinetics of the dHEV/wt progeny were comparable to those of C-rHEV and UC-rHEV, reaching titers exceeding 10^7^ copies/mL at 16 days post-infection (dpi) (Figure 6). These results indicated that progeny virions from plasmid-based HEV exhibited infectivity and replication efficiency comparable to those of in vitro-transcribed HEV RNA.

### 3.9. Effect of the Vaccinia Virus Capping Enzyme on HEV Replication

In cells transfected with dHEV/wt, HEV RNA production exhibited a temporal pattern similar to that of uncapped RNA (UC-rHEV) (Figure 2B). Analysis of the capping status of HEV RNA in the culture supernatant (Figure 3) suggested that viral genome capping did not occur efficiently at the early stage of infection within the cells. To enhance HEV genome capping intracellularly, expression plasmids for the vaccinia virus capping enzyme, pCAG-D1R and pCAG-D12L, were co-transfected with dHEV/wt, and HEV RNA titers were monitored over time. However, the presence of the vaccinia virus capping enzyme did not alter viral genome amplification in the culture medium (Figure 7). Additionally, the increase in the HEV RNA titer for dHEV/wt was slower compared that observed in Figure 2B. This delay was likely due to the reduced amount of dHEV/wt plasmid DNA used in the transfection to accommodate co-transfection with the vaccinia virus capping enzyme expression plasmid.

## 4. Discussion

In this study, we developed a single-step, plasmid-based reverse genetics system by inserting the cell-culture-adapted HEV genome (JE03-1760F_p13-3) between an HH ribozyme at the 5′ end and an HDV ribozyme at the 3′ end, under the control of a CMV-IE promoter. HEV derived from this plasmid successfully replicated within cells and was released into the culture supernatant at levels comparable to those observed with in vitro-transcribed RNA. Time-course analysis of HEV RNA titers in the culture supernatant revealed a slower increase in virus levels compared to those in in vitro-synthesized capped RNA-derived HEV, whereas the replication kinetics were comparable to those of uncapped RNA-derived HEV. This delay in viral replication was likely due to the slower capping speed of plasmid-derived HEV. Since HEV transcription from the plasmid is regulated by the HH and HDV ribozymes leading to the generation of full-length HEV genomes without a 5′ cap, we attempted to enhance the capping efficiency by co-transfecting cells with vaccinia virus capping enzyme expression plasmids. However, no significant increase in HEV RNA titers was observed, suggesting that the vaccinia virus capping enzyme was not functional in this system.

Reverse genetics systems for HEV have been previously established for genotypes other than HEV-2, including HEV-1 and HEV-3 to HEV-8 [42,52,53], all of which relied on RNA-based approaches. Additionally, RNA-based reverse genetics systems have been reported for rat HEV and avian HEV [54,55]. These RNA-based systems involved multiple steps, including HEV genome cloning, in vitro RNA synthesis, capping, and transfection, requiring meticulous handling due to RNA’s susceptibility for degradation. In contrast, a plasmid-based reverse genetics system simplifies this process by eliminating the need for in vitro RNA transcription and capping. Furthermore, the genetic marker mutation (NaeI restriction enzyme site) introduced in our plasmid system was retained in the progeny virus, demonstrating the feasibility of using this system for genetic modifications, such as generating mutant HEV strains.

For single-stranded positive-sense RNA viruses, plasmid-based reverse genetics systems serve as powerful research tools. Such systems have been developed for SARS-CoV-2 (coronavirus) [56], norovirus (picornavirus) [57,58], feline calicivirus [37], West Nile virus [59], Japanese encephalitis virus [60] (both flaviviruses), porcine epidemic diarrhea virus (alphacoronavirus) [61], and astrovirus [62]. However, reports on plasmid-based HEV reverse genetics systems remain limited, with only one prior study available [43].

The previously reported plasmid-based HEV reverse genetics system utilized T7 RNA polymerase-expressing cells and a T7 promoter-linked HEV-3 genome that lacked the HDV ribozyme following the polyA tail [43]. Additionally, a vaccinia virus capping enzyme expression vector was co-transfected to cap the HEV RNA genome [43]. Since self-cleaving sequences such as the HDV ribozyme were not incorporated, it was possible that the synthesized genome retained additional sequences downstream of the polyA tail. However, analysis of the produced viral genome and subsequent infection experiments indicated that any additional sequences were removed to an extent that did not affect viral replication [43]. Although T7 promoter-based systems are commonly employed [30,31,32,33], they require T7 RNA polymerase-expressing cells or expression constructs, which pose a limitation. Furthermore, the absence of self-cleaving sequences at the 3′ end may reduce initial viral replication and production efficiency. To overcome these limitations, our study employed a mammalian expression promoter and incorporated both HH and HDV ribozymes to facilitate the generation of full-length HEV genomes in a single-step process. However, the HEV genome synthesized by this system initially lacked a 5′ cap, necessitating an exogenous capping step via the introduction of a capping enzyme. Despite the introduction of the vaccinia virus capping enzyme expression plasmids, as described in previous reports [34,40,63,64], no effect on HEV replication was observed, indicating that enzymatic capping did not occur.

The RNA transcribed from the dHEV/wt plasmid was initially processed in the nucleus, where capping could occur upstream of the HH ribozyme. This RNA was subsequently cleaved by both HH and HDV ribozymes, resulting in an uncapped RNA genome in the cytoplasm. Therefore, it was anticipated that expressing the vaccinia virus capping enzyme in the cytoplasm would cap the HEV genome derived from the dHEV/wt plasmid. However, since the 5′ end of the RNA cleaved by the HH ribozyme lacked a phosphate group and instead possessed a 5′-OH [65], it was not a suitable substrate for the capping enzyme, thereby explaining the lack of capping despite the cytoplasmic expression of the vaccinia virus capping enzyme. To address this issue, an alternative strategy would involve aligning the transcription start site with the 5′ end of the HEV genome to utilize the nuclear capping system. This strategy would be advantageous, as would obviates the need for an exogenous capping mechanism. However, this approach would limit the use of regulatory sequences such as the TAR element, which was essential for enhancing expression in this study (Figure 1C). Further investigation is warranted to evaluate the feasibility and efficacy of this strategy in future studies.

## 5. Conclusions

We developed a plasmid-based, single-step HEV expression system incorporating HH and HDV ribozymes at the 5′ and 3′ ends of the HEV genome, respectively, under a CMV-IE promoter. The resulting viral yield was comparable to that of in vitro-synthesized capped HEV RNA, despite exhibiting production kinetics similar to uncapped HEV RNA. Viral proteins and negative-strand RNA were expressed at levels equivalent to those from capped HEV RNA. The introduced genetic marker was retained in the produced HEV genome, validating the feasibility of site-directed mutations. Our findings demonstrated that self-cleaving ribozymes facilitated full-length genome production, underscoring their utility for single-stranded positive-sense RNA virus studies. Notably, co-expression of the vaccinia virus capping enzyme expression plasmid did not enhance HEV production, necessitating further investigations to optimize efficiency.

## Figures and Tables

**Figure 1 viruses-17-00669-f001:**
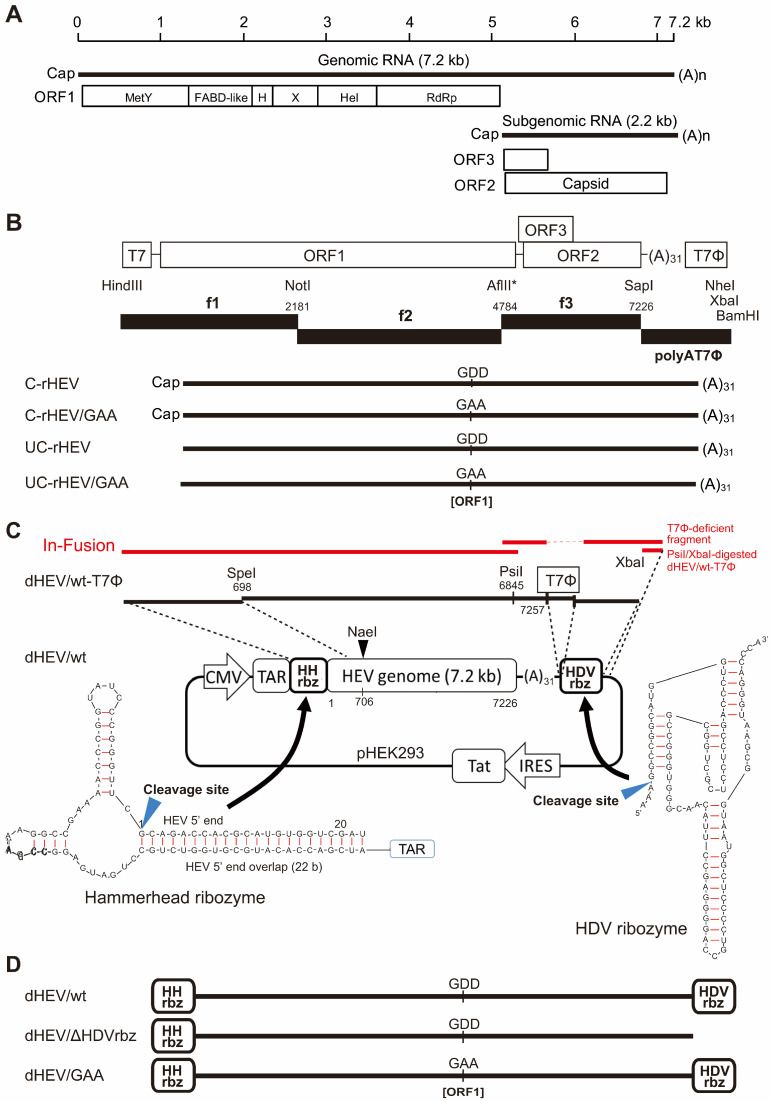
Schematic representation of the plasmid-based HEV expression vector. (**A**) Schematic diagram illustrating the HEV genomic and subgenomic RNA structures. The HEV genome consists of three open reading frames (ORFs): ORF1 encodes a nonstructural polyprotein with multiple functional domains, including a unique MetY domain, an FABD-like domain, a hypervariable region (H), an X domain (macrodomain), a helicase (Hel), and an RNA-dependent RNA polymerase (RdRp). ORF2 encodes the capsid protein, while ORF3 encodes a small phosphoprotein involved in viral egress. (**B**) Construction of the pJE03-1760F_p13-3 HEV genome and in vitro transcription of capped (C-rHEV) and uncapped (UC-rHEV) HEV RNA. The full-length genomic sequence is cloned downstream of the T7 promoter (T7), with transcription terminated at the T7 terminator (T7Φ). C-rHEV/GAA and UC-rHEV/GAA are replication-deficient negative control constructs, in which the GDD motif at the RdRp catalytic site is mutated to GAA. A-to-C substitution at nt 4707 and 4710 result in D-to-A amino acid mutations at positions 1561 and 1562 in the RdRp domain. (**C**) Schematic representation of the construction of the plasmid-based HEV expression vector (dHEV/wt). The HEV genome is inserted downstream of the transactivation response (TAR) element in the pHEK293 plasmid and flanked by hammerhead (HH) and hepatitis delta virus (HDV) ribozymes at the 5′ and 3′ ends, respectively, to ensure precise cleavage and the generation of authentic HEV RNA termini. The dHEV/wt-T7ϕ plasmid is represented by black lines, while the dHEV/wt plasmid, which lacks the T7 terminator and is generated via In-Fusion cloning, is depicted in red. (**D**) Schematic overview of the plasmid-based HEV constructs used in this study. dHEV/ΔHDVrbz is a construct lacking the 3′ HDV ribozyme in dHEV/wt. dHEV/GAA is a replication-deficient negative control construct, in which the GDD motif at the RdRp catalytic site is mutated to GAA. A-to-C substitution at nt 4707 and 4710 result in D-to-A amino acid mutations at positions 1561 and 1562 in the RdRp domain.

**Figure 2 viruses-17-00669-f002:**
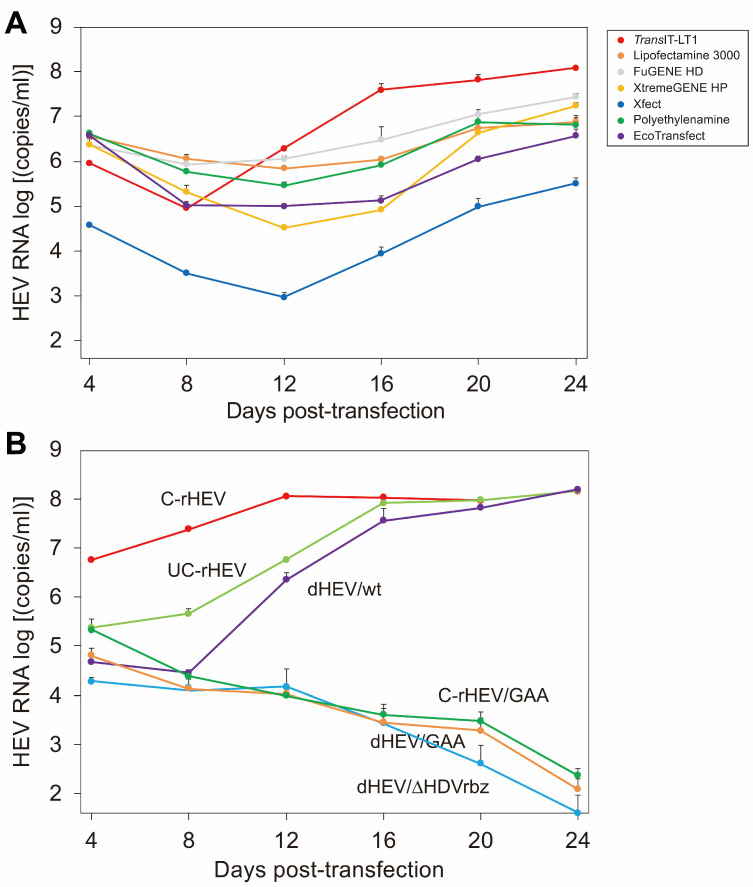
Comparison of HEV production efficiency between plasmid-based HEV expression systems and in vitro-transcribed HEV RNA genomes. (**A**) Evaluation of seven transfection reagents for their efficiency in facilitating the replication of plasmid-derived HEV (dHEV/wt). The reagents tested include *Trans*IT-LT1, Lipofectamine 3000, FuGENE HD, X-treamGENE HP, Xfect, polyethylenimine, and Eco Transfect. (**B**) HEV growth efficiency following transfection with dHEV/wt, its polymerase-deficient mutant (dHEV/GAA; negative control), a construct lacking the HDV ribozyme (dHEV/ΔHDVrbz), and in vitro-transcribed capped (C-rHEV, C-rHEV/GAA) or uncapped (UC-rHEV) HEV RNA genomes. HEV RNA levels in the culture supernatants of PLC/PRF/5 cells are quantified over a period of 24 days. Due to inherent technical limitations, the carryover of transfected plasmid DNA or in vitro-transcribed RNA during nucleic acid extraction is, to some extent, unavoidable. To address this, negative controls (dHEV/GAA for the dHEV/wt construct and C-rHEV/GAA for the C-rHEV and UC-rHEV constructs) are included. These controls represent the signal levels attributable to DNA or RNA carryover, and serve as a benchmark for interpretation. Data are presented as the mean ± standard deviation (SD) from three independent experiments (*n* = 3).

**Figure 3 viruses-17-00669-f003:**
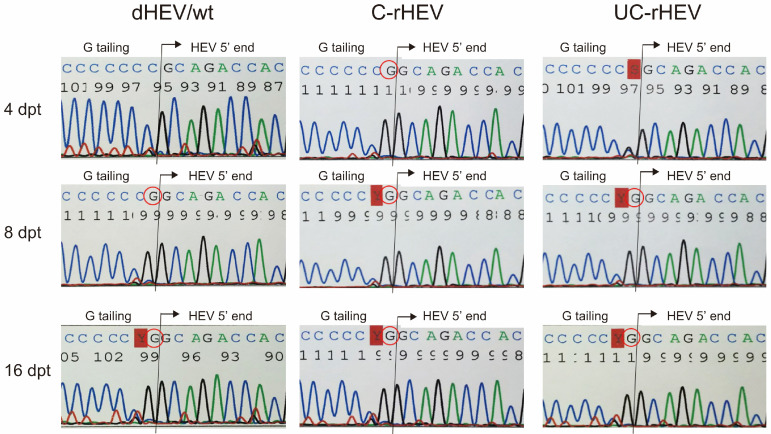
Delayed 5′ capping of HEV RNA in PLC/PRF/5 cells transfected with HEV plasmid DNA or uncapped HEV RNA, compared to capped HEV RNA. PLC/PRF/5 cells were transfected with either the dHEV/wt plasmid or in vitro-transcribed capped (C-rHEV) or uncapped (UC-rHEV) HEV RNA. RNA were extracted from the culture supernatant collected at 4, 8, and 16 dpt, and subjected to 5′ rapid amplification of cDNA ends (5′-RACE) followed by direct sequencing. The circled “G” denotes the presence of the 5′ 7-methylguanosine (m7G) cap structure. Red boxes indicate nucleotides with uncertain identity due to a low signal intensity in the electropherogram. The results were reproducible; representative electropherograms from two independent experiments are shown.

**Figure 4 viruses-17-00669-f004:**
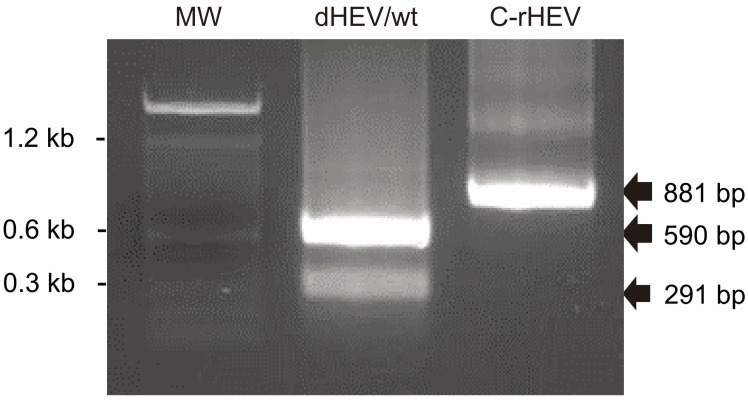
Retention of a genetic marker in the plasmid-derived HEV genome. HEV particle-associated RNA genomes obtained from culture supernatants at 20 dpt were analyzed via reverse transcription PCR and digested with NaeI, which was introduced as a genetic marker. Abbreviations: MW, molecular weight marker; dHEV/wt, plasmid-derived HEV expression vector; C-rHEV, in vitro-transcribed capped HEV RNA.

**Figure 5 viruses-17-00669-f005:**
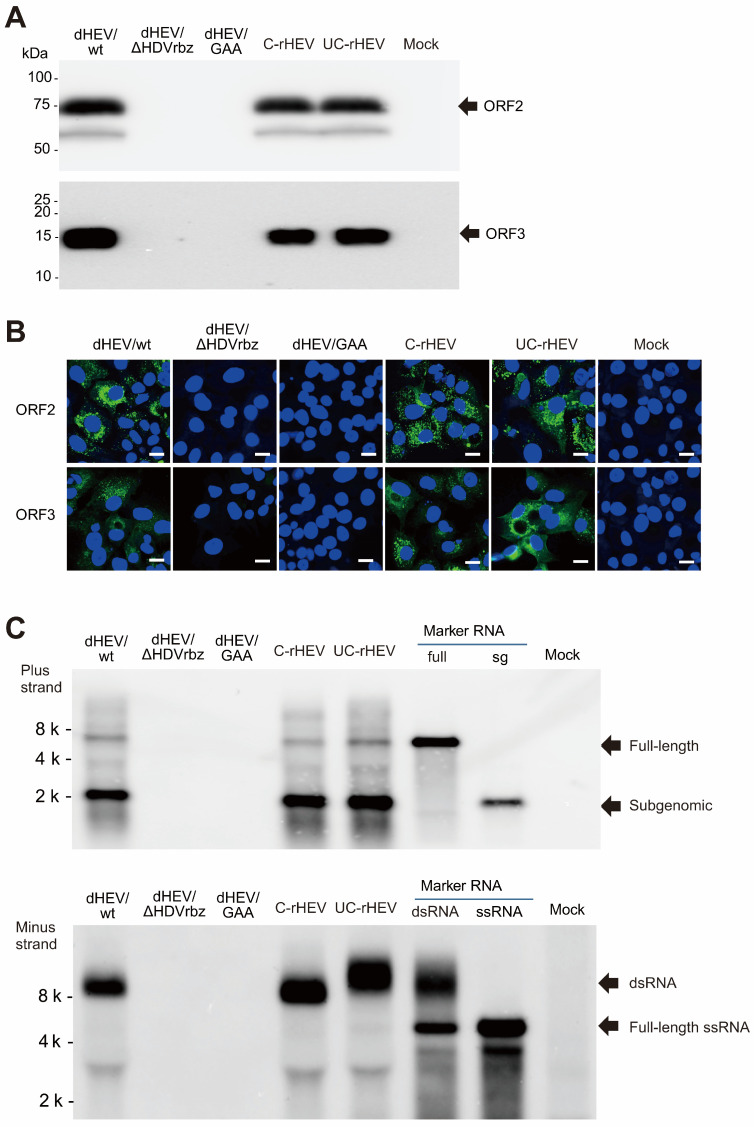
Comparison of ORF2 and ORF3 expression and HEV RNA replication between plasmid-derived and in vitro-transcribed HEV. (**A**) Western blot analysis of viral proteins in culture supernatants from PLC/PRF/5 cells transfected with either the HEV expression plasmid DNA or in vitro-transcribed HEV RNA at 20 dpt. Each lane was loaded with 5 μL of culture supernatant (without concentration). ORF2 and ORF3 proteins were detected using anti-HEV ORF2 monoclonal antibody (MAb) (**upper panel**) or anti-anti-HEV ORF3 MAb (**lower panel**). (**B**) Immunofluorescence analysis of ORF2 and ORF3 protein expression in PLC/PRF/5 cells transfected with the indicated HEV plasmids or in vitro-transcribed HEV RNA at 24 dpt. Cells were stained with an anti-HEV ORF2 (**upper panel**) or anti-anti-HEV ORF3 (**lower panel**) MAbs, followed by Alexa Fluor 488-conjugated anti-mouse IgG. Nuclei were counterstained with DAPI. Bar, 20 μm. (**C**) Northern blot analysis of intracellular HEV RNA genomes using an antisense RNA probe (Plus strand detection) or a sense RNA probe (minus strand detection) at 24 dpt. As positive controls (marker RNA), positive-strand full-length (full) and subgenomic (sg) positive-strand HEV RNA (1 ng/lane) were used (**upper panel**), along with double-stranded full-length HEV RNA (dsRNA, 100 pg/lane) and single-stranded full-length negative-strand HEV RNA (ssRNA, 50 pg/lane) were used (**lower panel**). Abbreviations: dHEV/wt, plasmid-based HEV expression construct; dHEV/ΔHDVrbz, plasmid-based HEV construct lacking the HDV ribozyme; dHEV/GAA, HEV construct with the GAA mutation; C-rHEV; capped in vitro-transcribed HEV RNA; UC-rHEV; uncapped in vitro-transcribed HEV RNA.

**Figure 6 viruses-17-00669-f006:**
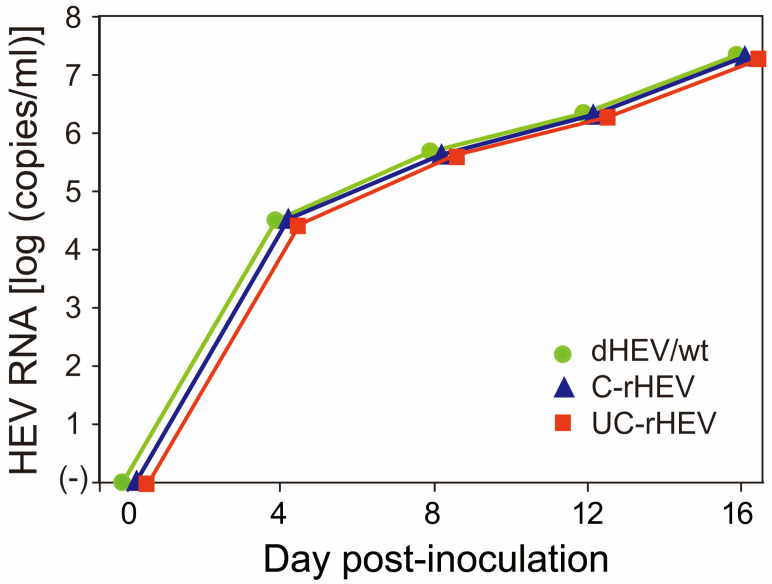
Comparison of HEV growth efficiency between plasmid-derived and RNA-derived HEV. A549 cells were inoculated with culture supernatants (1 × 10^5^ copies/well) collected from PLC/PRF/5 cells transfected with either the dHEV/wt plasmid, the in vitro-transcribed capped HEV RNA (C-rHEV), or the in vitro-transcribed uncapped HEV RNA (UC-rHEV). HEV RNA levels in the culture supernatants of A549 cells were quantified over a 16-day period. Data are presented as the mean ± standard deviation (SD) from three independent experiments (*n* = 3). Error bars are not visible due to minimal variation among replicates.

**Figure 7 viruses-17-00669-f007:**
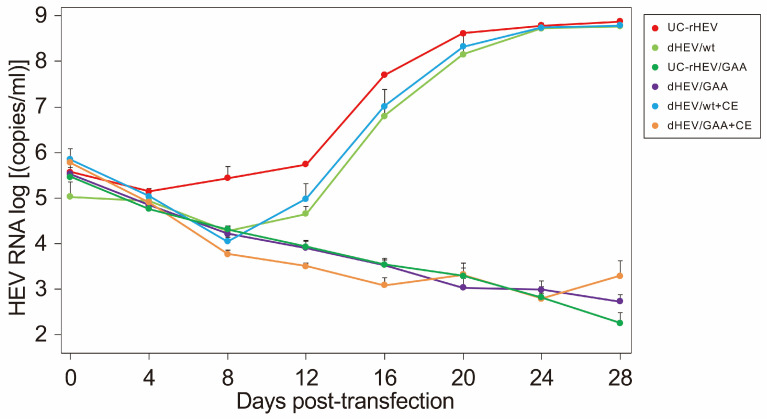
Replication of plasmid-derived HEV in the presence or absence of the vaccinia virus capping enzyme. PLC/PRF/5 cells were transfected with dHEV/wt or dHEV/GAA plasmid alone or co-transfected with vaccinia virus capping enzyme expression plasmids (dHEV/wt+CE or dHEV/GAA+CE), in comparison to in vitro-transcribed uncapped HEV RNA genome (UC-rHEV). HEV replication efficiency was assessed by quantifying HEV RNA in culture supernatants over a 28-day period. Due to inherent technical limitations, the carryover of transfected plasmid DNA or in vitro-transcribed RNA during nucleic acid extraction was, to some extent, unavoidable. To address this, negative controls (dHEV/GAA for dHEV/wt construct and UC-rHEV/GAA for UC-rHEV construct) were included. These controls represented signal levels attributable to DNA or RNA carryover, and served as a benchmark for interpretation. Data are presented as the mean ± standard deviation (SD) from three independent experiments (*n* = 3).

## Data Availability

All data are presented in the manuscript.

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
