# Peer review of "Plasmid-Based Reverse Genetics System Enabling One-Step Generation of Genotype 3 Hepatitis E Virus"

_viruses, 2025, doi:10.3390/v17050669_

Round 1
Reviewer 1 Report
Comments and Suggestions for Authors
This manuscript describes the development of a DNA-launch reverse genetics system for hepatitis E virus (HEV). Such a system will be useful for HEV study and avoid the hassles of in vitro transcription and capping of the RNA-based systems that are widely used in the HEV field. The results are promising and show that it performs similarly to RNA transfection although with some delay. However, there are serious concerns with the data presented. Repeated experiments are needed to confirm the findings.
Major concerns:
- There are no error bars or statistical analysis of the data in multiple figures (Fig. 2, 6 and 7). The results shown are mean values from two or three wells for each time points. At least three experiments should be done.
- In Figure 2A, the HEV RNA level should increase from 4 to 8 dpt and from 8 to 12 dpt, based on Fig 2B. This also indicates the importance of repeating the experiments three times.
- In Figure 2B, at 0 dpt, the over 6 logs of RNA in culture supernatant from the transfection with plasmid DNA of dHEV/wt and its mutants appear impossible. This indicates DNA carryover during RNA isolation. When doing qPCR, include a control of no reverse transcription to confirm there is no DNA carryover. Also, for the RNA transfection, it is also hard to achieve the high HEV RNA level in culture supernatant at 0 dpt. This indicates RNA carryover from transfection. The same concern is also for Figure 7. Repeat the experiments with proper controls.
- In Figure 5A, Western blotting shows the detection of ORF2 and ORF3 products of cell culture supernatant samples at 20 dpt, how much was loaded into the gel? Were concentrated samples used? Detect actin of the same blot to show the absence of detached cells. Were there dead cells? How about cell lysate? This should be done too.
- In Figure 5C, Is this also from 20 dpt samples? Add to the legend. Why is the full-length RNA much less than the subgenomic? A qPCR should be run to confirm the difference between the full length and subgenomic RNA using primers of ORF1 for full length and primers of ORF2 for subgenomic RNA. In the lower panel, why is the UC-rHEV dsRNA band higher than C-rHEV and dHEV/wt lanes as well as dsRNA marker?
- In Figure 6, did you include the virus from the UC-rHEV in the experiment?
Minor concerns
- Line 45, delete “family” after Hepeviridae.
- Line 51, Y domain is considered an extension of Met domain. Update.
- Line 58, rewrite to state it is present in quasi-enveloped but not non-enveloped particles.
- Line 62, rewrite this sentence. There are eight genotypes in the species Paslahepevirus balayani.
- Line 64, update this sentence. HEV-3 and HEV-4 are prevalent in many developing countries too due to their zoonotic feature.
- In Figure 3, the red square covers the symbol inside and makes it less visible. Change to another color. Also, add explanation of the symbols in legend.
- In Figure 5B, add size bars to the images.
Comments on the Quality of English Language
Improvement is needed. See comments above.
Author Response
Responses to the comments from Reviewer 1
Comments and Suggestions for Authors
This manuscript describes the development of a DNA-launch reverse genetics system for hepatitis E virus (HEV). Such a system will be useful for HEV study and avoid the hassles of in vitro transcription and capping of the RNA-based systems that are widely used in the HEV field. The results are promising and show that it performs similarly to RNA transfection although with some delay. However, there are serious concerns with the data presented. Repeated experiments are needed to confirm the findings.
 Response: We sincerely thank you for your insightful comments and constructive suggestions, which have helped us improve the quality of our manuscript. In response, we have thoroughly revised the manuscript and the figures as detailed below.
Major concerns:
Comment 1: There are no error bars or statistical analysis of the data in multiple figures (Fig. 2, 6 and 7). The results shown are mean values from two or three wells for each time points. At least three experiments should be done.
Response: We appreciate your important comment. We have repeated the experiments and revised Figures 2, 6, and 7 accordingly. The updated figures now present data as the mean ± standard deviation (SD) from three independent experiments (n = 3).
Comment 2: In Figure 2A, the HEV RNA level should increase from 4 to 8 dpt and from 8 to 12 dpt, based on Fig 2B. This also indicates the importance of repeating the experiments three times.
Response: As noted above in response to Major Comment 1, Figures 2A and 2B have been updated with data from three independent experiments (n = 3), presented as mean ± SD. In the revised Figure 2A, HEV RNA levels following dHEV/wt transfection with TransIT-LT1 show a clear increase from 8 to 12 dpt, in line with Figure 2B. Therefore, we believe there is no discrepancy in the kinetics of HEV RNA production between the two panels.
Comment 3: In Figure 2B, at 0 dpt, the over 6 logs of RNA in culture supernatant from the transfection with plasmid DNA of dHEV/wt and its mutants appear impossible. This indicates DNA carryover during RNA isolation. When doing qPCR, include a control of no reverse transcription to confirm there is no DNA carryover. Also, for the RNA transfection, it is also hard to achieve the high HEV RNA level in culture supernatant at 0 dpt. This indicates RNA carryover from transfection. The same concern is also for Figure 7. Repeat the experiments with proper controls.
Response: We acknowledge your concern. Due to inherent technical limitations, carryover of transfected plasmid DNA or in vitro-transcribed RNA during nucleic acid extraction is, to some extent, unavoidable. To address this, we included negative controls (dHEV/GAA for dHEV constructs and C-rHEV/GAA or UC-rHEV/GAA for C-rHEV and UC-rHEV constructs) in all relevant experiments. These controls represent signal levels attributable to DNA or RNA carryover, and serve as a benchmark for interpretation. This approach is consistent with practices in previous studies in the field.
Comment 4: In Figure 5A, Western blotting shows the detection of ORF2 and ORF3 products of cell culture supernatant samples at 20 dpt, how much was loaded into the gel? Were concentrated samples used? Detect actin of the same blot to show the absence of detached cells. Were there dead cells? How about cell lysate? This should be done too.
Response: We have clarified in the revised figure legend that 5 µL of culture supernatant (without concentration) was loaded per lane. As noted in the legend to Figure 2 and based on our extensive experience with PLC/PRF/5 cells, no detached or dead cells were observed over the course of 24–28 days of culture (PMIDs: 17325363, 18620009, 19141456, 19369433, 20107086, 22048607, 23041252, 27485920, and 31362004). Therefore, we believe the inclusion of actin as a loading control is not essential. Furthermore, the detection of ORF2 and ORF3 proteins by immunofluorescence in transfected cells (Figure 2B) supports the presence of intracellular expression. Given that IFA provides both qualitative and spatial information on viral protein expression, we respectfully consider that additional Western blotting of cell lysates is not necessary.
Comment 5: In Figure 5C, Is this also from 20 dpt samples? Add to the legend. Why is the full-length RNA much less than the subgenomic? A qPCR should be run to confirm the difference between the full length and subgenomic RNA using primers of ORF1 for full length and primers of ORF2 for subgenomic RNA. In the lower panel, why is the UC-rHEV dsRNA band higher than C-rHEV and dHEV/wt lanes as well as dsRNA marker?
Response: As noted in the figure legend, the samples used were collected at 24 dpt. HEV subgenomic RNA, which encodes the highly expressed ORF2 and ORF3 proteins, is known to accumulate to higher levels than full-length genomic RNA. This finding is consistent with previous reports (PMIDs: 27485920 and 29471051). The objective of this Northern blotting experiment was to demonstrate that the dHEV/wt construct is capable of producing both full-length and subgenomic RNA transcripts, similar to C-rHEV. While quantitative RT-PCR could provide additional information, we believe it is not essential for the purpose of this figure. Regarding the apparent upward shift of the dsRNA band in the UC-rHEV lane, we believe this to be a gel artifact, as similar upward shifts were observed in nonspecific bands in the same lane.
Comment 6: In Figure 6, did you include the virus from the UC-rHEV in the experiment?
Response: Yes, the virus derived from the UC-rHEV construct has been included in the updated Figure 6.
Minor concerns
Comment 1: Line 45, delete “family” after Hepeviridae.
Response: Revised as suggested (Line 50).
Comment 2: Line 51, Y domain is considered an extension of Met domain. Update.
Response: The sentence has been updated to:
“ORF1 encodes a non-structural polyprotein essential for viral replication, which contains several functional domains, including a unique MetY domain that combines the previously described methyltransferase (Met) and Y domains, a fatty acid binding domain (FABD)-like domain, hypervariable region, X (macro) domain, helicase (Hel), and RNA-dependent RNA polymerase (RdRp) [16-18].” (Lines 55-59)
Comment 3: Line 58, rewrite to state it is present in quasi-enveloped but not non-enveloped particles.
Response: The sentence has been revised to clarify that ORF3 is present in quasi-enveloped HEV particles (Lines 64-65).
Comment 4: Line 62, rewrite this sentence. There are eight genotypes in the species Paslahepevirus balayani.
Response: Revised to:
“Within the species Paslahepevirus balayani, eight genotypes of HEV have been identified, designated HEV-1 through HEV-8.” (Lines 65-66)
Comment 5: Line 64, update this sentence. HEV-3 and HEV-4 are prevalent in many developing countries too due to their zoonotic feature.
Response: Updated to:
“HEV-3 and HEV-4 are prevalent not only in industrialized countries but also in many developing countries due to their zoonotic nature, and they may lead to chronic infection, especially in immunocompromised individuals [24].” (Lines 70-73)
Comment 6: In Figure 3, the red square covers the symbol inside and makes it less visible. Change to another color. Also, add explanation of the symbols in legend.
Response: As noted in the legend, the red squares denote “undetermined” nucleotides due to low signal intensity. These annotations are automatically generated by the sequencer software and cannot be modified or recolored.
Comment 7: In Figure 5B, add size bars to the images.
Response: A scale bar of 20 µm has been added to the figure.
Comments on the Quality of English Language
Improvement is needed. See comments above.
Response: The manuscript has been carefully reviewed by an experienced medical editor whose first language is English.
We hope that these revisions adequately address your concerns and enhance the clarity and robustness of our manuscript. Once again, we sincerely thank you for your careful evaluation and helpful feedback.

Reviewer 2 Report
Comments and Suggestions for Authors
In the present manuscript, the authors described the development of a plasmid-based reverse genetic system to study hepatitis E virus (HEV). This paper is of interest for the field as HEV strains do not replicate efficiently in cell culture and alternative straightforward systems are needed to study the virus. The manuscript is clear and well written. Here are some points for consideration:
- Line 393/498/518: 2 or 3 wells from the same experiment? Were independent experiments performed to check that the results obtained are reproducible?
- Figure 3: Is a specific sequencing protocol used to identify capped genome? Were individual HEV genomes sequenced for each condition (a mixture of capped and uncapped genome could be present in the supernatant?)? How many genomes/samples were sequenced per condition? What does the nucleotide coloured in red refers to?
- Figure 7: Have you checked that the vaccinia virus capping enzyme was well expressed and functional in the system used?
- Line 576-577: Why was the vaccinia virus capping enzyme tested in this study if it was already expected that it would not work?
Minor comments:
- Line 40-41: individuals is repeated
- Line 52: New evidence suggests that the region of ORF1 first identified as a “PCP domain” does not encode a protease and is not a PCP domain.
- Line 56: Three forms of ORF2 have been reported. Moreover, ORF2c and ORFs should be defined if they are mentioned in the text.
- Line 89: “to address this” : What does “this” refer to?
- Line 349-368: this paragraph could be more concise as well described in the materials and method section.
- Figure 2: Did you treat supernatants with DNase before RNA extraction to remove residual DNA plasmid? Is it possible to monitor intracellular replication using a plasmid DNA system or there is too much contamination from the plasmid DNA transfected (even after DNase treatment?) when performing the RT-qPCR?
- Figure 2B: Could the use of different transfection protocols between RNA and DNA constructs also explain the difference in kinetics between C-rHEV and dHEV/wt?
- Line 386: The use of “Comparable amplification efficiency” is not clear.
- Line 583: TAR is not defined in the text (abbreviation). Why is this element important in this study to enhance expression?
Author Response
Responses to the comments from Reviewer 2
Comments and Suggestions for Authors
In the present manuscript, the authors described the development of a plasmid-based reverse genetic system to study hepatitis E virus (HEV). This paper is of interest for the field as HEV strains do not replicate efficiently in cell culture and alternative straightforward systems are needed to study the virus. The manuscript is clear and well written. Here are some points for consideration:
Response: We sincerely appreciate your positive evaluation and valuable comments, which have significantly contributed to the improvement of our manuscript.
Line 393/498/518: 2 or 3 wells from the same experiment? Were independent experiments performed to check that the results obtained are reproducible?
Response: We appreciate your insightful comment regarding experimental reproducibility. We have repeated the experiments and revised Figures 2, 6, and 7 accordingly. The updated figures now present data as the mean ± standard deviation (SD) from three independent experiments (n = 3), thereby confirming the reproducibility of the results.
Figure 3: Is a specific sequencing protocol used to identify capped genome? Were individual HEV genomes sequenced for each condition (a mixture of capped and uncapped genome could be present in the supernatant?)? How many genomes/samples were sequenced per condition? What does the nucleotide coloured in red refers to?
Response: Thank you for your important question. HEV RNA extracted from the supernatant was reverse-transcribed, amplified by PCR, and subjected to direct sequencing. As a result, the electropherograms reflect a mixture of capped and uncapped genomes. Red boxes mark positions with ambiguous base calls due to low signal intensity. For example, at 4 dpt in the UC-rHEV sample, the nucleotide (located upstream of the extreme 5’ end nucleotide) is designated as 'S', indicating a mixture of G and C bases, suggesting the coexistence of capped and uncapped genomes. In other five samples, the red-highlighted nucleotides (located upstream of the circled “G”) were consistently undetermined and are likely due to background noise inherent to the sequencing process. Representative electropherograms from two independent experiments are shown, and the results were reproducible.
Figure 7: Have you checked that the vaccinia virus capping enzyme was well expressed and functional in the system used?
Response: We appreciate your pertinent concern regarding the confirmation of capping enzyme expression. We agree that direct confirmation would be ideal; however, due to the unavailability of specific antibodies, and the risk of activity loss upon epitope tagging, we were unable to assess protein expression directly. Nevertheless, the VVCE expression plasmid used in this study is identical to those validated in previous studies (PMIDs: 32106549 and 32818718). Furthermore, we employed strong mammalian promoters and optimized transfection reagents, suggesting efficient expression. While the data cannot be shown in this manuscript, the functionality of this construct has been confirmed in a separate study currently under preparation. We appreciate your understanding of these technical limitations.
Line 576-577: Why was the vaccinia virus capping enzyme tested in this study if it was already expected that it would not work?
Response: Thank you for your thoughtful comment. As discussed in the manuscript, a previous study utilizing a plasmid-based HEV reverse genetics system co-transfected a VVCE expression vector to achieve RNA capping [41]. However, the expected results were not obtained. In our study, we investigated this further, and the Discussion section elaborates on the reasons why VVCE was ineffective in this context (Lines 976-1028).
Minor comments:
Line 40-41: individuals is repeated
Response: Thank you for pointing this out. The repeated word has been removed as suggested.
Line 52: New evidence suggests that the region of ORF1 first identified as a “PCP domain” does not encode a protease and is not a PCP domain.
Response: We have revised the description and updated Figure 1A accordingly. The sentence now reads:
“ORF1 encodes a non-structural polyprotein essential for viral replication, which contains several functional domains, including a unique MetY domain that combines the previously described methyltransferase (Met) and Y domains, a fatty acid binding domain (FABD)-like domain, hypervariable region, X or macro domain, helicase (Hel), and RNA-dependent RNA polymerase (RdRp) [16-18].” (Lines 55-59)
Line 56: Three forms of ORF2 have been reported. Moreover, ORF2c and ORFs should be defined if they are mentioned in the text.
Response: The sentence has been revised as follows:
“The ORF2 protein exists in infectious, glycosylated, and cleaved forms. The glycosylated and cleaved forms are considered to function as immunogenic decoys [19,20].” (Lines 62-64)
Line 89: “to address this” : What does “this” refer to?
Response: We revised the sentence for clarity:
“To address the issue of unwanted extraneous sequences at the 3′ end of the viral genome, a strategy employing the HDV ribozyme has been developed. This ribozyme enables autocatalytic cleavage at the 3′ end of the RNA transcript, thereby producing a precise viral genome terminus [38,49,41].” (Lines 312-315)
Line 349-368: this paragraph could be more concise as well described in the materials and method section.
Response: We have condensed the paragraph for conciseness and consistency with the Methods section. The revised version is:
“To further establish a plasmid-based reverse genetics system for HEV, we constructed the dHEV/wt plasmid (Figure 1C) using pJE03-1760F_p13-3 as the template. The pHEK293 vector was first modified to include additional restriction sites, after which the HDV ribozyme was inserted via NheI/XbaI cloning. The 3´-terminal HEV genomic fragment (nt 698–7226), containing a 31-nt polyA tail and a T7 terminator, was excised from pJE03-1760F_p13-3 and ligated into the modified vector. Subsequently, the 5´-terminal HEV fragment (nt 1–698), fused to an HH ribozyme, was PCR-amplified and inserted upstream of the 3´ fragment. To eliminate the T7 terminator, a replacement fragment was introduced by In-Fusion cloning, enabling direct linkage between the polyA sequence and the HDV ribozyme. Silent mutations introducing a NaeI restriction site were incorporated at nt 706 and 709 as a genetic marker. The assembled dHEV/wt plasmid was verified by sequencing.” (Lines 635-646)
Figure 2: Did you treat supernatants with DNase before RNA extraction to remove residual DNA plasmid? Is it possible to monitor intracellular replication using a plasmid DNA system or there is too much contamination from the plasmid DNA transfected (even after DNase treatment?) when performing the RT-qPCR?
Response: Thank you for raising this important point. Serum contains endogenous DNases; however, plasmid DNA complexed with transfection reagents may resist degradation. Complete elimination of carryover DNA or RNA is technically challenging. To mitigate this, we included negative controls (dHEV/GAA and C-rHEV/GAA or UC-rHEV/GAA) in all RT-qPCR assays to account for non-replicative background signals. This strategy is consistent with previous studies in the field and allows for accurate interpretation of replication-specific signals.
Figure 2B: Could the use of different transfection protocols between RNA and DNA constructs also explain the difference in kinetics between C-rHEV and dHEV/wt?
Response: We appreciate this observation. Although equal mass (500 ng) of RNA or DNA was transfected, their molar quantities differ, and the amount of RNA generated from the plasmid is not precisely known. As such, the differences in transfection efficiency may indeed contribute to the observed kinetic differences between C-rHEV and dHEV/wt. Nevertheless, as shown in Figure 3, the delay in capping appears to be the predominant factor affecting viral kinetics.
Line 386: The use of “Comparable amplification efficiency” is not clear.
Response: To improve clarity, we revised the sentence to:
“Comparison of HEV production efficiency between plasmid-based HEV expression systems and in vitro-transcribed HEV RNA genomes.” (Lines 725-726)
Line 583: TAR is not defined in the text (abbreviation). Why is this element important in this study to enhance expression?
Response: Thank you for pointing this out. We have added the following explanation to the revised manuscript:
“The pHEK293 Ultra Expression Vector I was purchased from TaKaRa Bio (TaKaRa Bio Inc., Shiga, Japan). This vector, used as the backbone for dHEV/wt, contains a transactivation response (TAR) element and enables high-level recombinant protein expression via a transcriptional activation mechanism derived from HIV-1 TAR-Tat system, under the control of a cytomegalovirus (CMV) promoter. The TAR element is an RNA sequence originating from HIV-1 that forms a stem-loop structure at the 5' end of the viral RNA. The Tat (transactivator of transcription) protein is an RNA binding transactivator that specifically interacts with the TAR element to enhance transcriptional activity.” (Lines 345-352)

Reviewer 3 Report
Comments and Suggestions for Authors
Title: A Plasmid-Based Reverse Genetics System Enabling One-Step Generation of Genotype 3 Hepatitis E Virus
HEV is an important emerging but understudied zoonotic virus. Reverse genetic system is a powerful tool to study the molecular mechanism in virology. In the HEV field, all efficient reverse genetic platforms are RNA based, requiring in vitro transcription. A platform which uses plasmid and transfect into cell directly would greatly facilitate the fundamental research of HEV. This study developed a plasmid based HEV rescue system that enables direct intracellular transcription of the HEV genome under a CMV promoter. Even though the developed platform in this study is not as good as expected, the work is significant. The manuscript is also well written. Some minor concerns from this reviewer as below.
- Line 62, “HEV has eight identified genotypes, HEV-1 to HEV-8”, need to add description of “in the genus Paslahepevirus”.
- Line 106. Better to delete “as the HEV strain”
- Line 386, the title of figure 2 is not precise
- Line 409-411, the description looks not correct to figure 2B, HDV element should improve not “inhibit” HEV replication.
- Line 412, the title of 3.4 is a little bit confusing, it should be capping of plasmid-derived HEV genomic RNA.
- For figure 3, please indicate the meaning of red boxes in graphic.
- Suggest combining the result 3.5 figure 4 into figure 2.
- For figure 5C northern blot, the lanes of dHEV/ΔHDVrbz and dHEV/GAA looks over clean. According to the figure 2, there were still small amount of HEV genomic RNAs or subgenomic RNAs in cells.
Author Response
Responses to the comments from Reviewer 3
HEV is an important emerging but understudied zoonotic virus. Reverse genetic system is a powerful tool to study the molecular mechanism in virology. In the HEV field, all efficient reverse genetic platforms are RNA based, requiring in vitro transcription. A platform which uses plasmid and transfect into cell directly would greatly facilitate the fundamental research of HEV. This study developed a plasmid based HEV rescue system that enables direct intracellular transcription of the HEV genome under a CMV promoter. Even though the developed platform in this study is not as good as expected, the work is significant. The manuscript is also well written. Some minor concerns from this reviewer as below.
Response: Thank you very much for your kind evaluation and constructive comments, which have helped us improve the quality of our manuscript. Please find our detailed responses to your suggestions below.
Comment 1 (Line 62): “HEV has eight identified genotypes, HEV-1 to HEV-8”, need to add description of “in the genus Paslahepevirus”.
Response: We appreciate your suggestions. The sentence has been revised to: “Within the species Paslahepevirus balayani, eight genotypes of HEV have been identified, designated HEV-1 through HEV-8.” (Lines 65-66)
Comment 2 (Line 106): Better to delete “as the HEV strain”.
Response: Thank you for your suggestion. The phrase “as the HEV strain” has been removed to improve clarity.
Comment 3 (Line 386): the title of figure 2 is not precise
Response: We agree with your comment and have revised the figure title to:
“Comparison of HEV production efficiency between plasmid-based HEV expression systems and in vitro-transcribed HEV RNA genomes.” (Lines 725-726)
Comment 4 (Line 409-411): the description looks not correct to figure 2B, HDV element should improve not “inhibit” HEV replication.
Response: Thank you for pointing this out. The description has been revised for accuracy:
“These findings suggest that the presence of extraneous plasmid-derived sequences downstream of the poly(A) tail at the 3' end of the HEV genome inhibits intracellular amplification of the HEV genome.” (Lines 772-774)
Comment 5 (Line 412): the title of 3.4 is a little bit confusing, it should be capping of plasmid-derived HEV genomic RNA.
Response: We appreciate suggestion. The title of Section 3.4 has been revised to: “Delayed Capping of Plasmid-Derived HEV genomic RNA in Cells”. (Line 775)
Comment 6 (Figure 3): please indicate the meaning of red boxes in graphic.
Response: Thank you for your comment. We have added a description in the figure indicating that the red boxes represent nucleotides with uncertain identity due to low signal intensity in the electropherogram. (Lines 792-794)
Comment 7: Suggest combining the result 3.5 figure 4 into figure 2.
Response: We understand your suggestion; however, Figure 2 presents data comparing HEV production efficiencies between different expression systems, while Figure 4 focuses on the detectability of a genetic marker in plasmid-derived HEV genomes. Since two figures illustrate distinct aspects of the study, we believe it is more appropriate to present them separately to avoid confusion.
Comment 8 (Figure 5C): northern blot lanes for dHEV/ΔHDVrbz and dHEV/GAA looks over clean. According to the figure 2, there were still small amount of HEV genomic RNAs or subgenomic RNAs in cells.
Response: You are correct in noting that small amounts of HEV genomic and subgenomic RNAs were detected by RT-qPCR at 24 dpt. However, due to the lower sensitivity of Northern blotting compared to RT-qPCR, we believe that these low-abundance RNA species were below the detection limit in the Northern blot analysis, resulting in the absence of visible signals for the dHEV/GAA and dHEV/ΔHDVrbz samples at 24 dpt.

Round 2
Reviewer 1 Report
Comments and Suggestions for Authors
The authors made revisions to address the reviewers’ comments. The majority of concerns were addressed, but some major ones remain.
- In Figure 2B, the 0-day data is misleading due to DNA carryover. You can subtract the value from the No-RT control. Or remove the 0-day time point. The RNA level for dHEV/wt in Fig 2A reduces almost a log from 4 to 8 dpt but a slight decrease in Fig 2B in the same time frame (increase in the old 2B). It seems the error bars do not reflect the variations of the new Fig2 with the old Fig2. An example is that the UC-rHEV and dHEV/wt lines were closer in the old 2B, but a much bigger difference in the new 2B. Explain if the data from the old figure was included, and if the new figure 2 uses all new data from three new experiments. Considering the time after the previous review, it is hard to get three new experiments (24 days for one experiment) done.
- It is essential to include actin control, as it is impossible to have no dead cells for cultures of 24-28 days. You can easily probe the same membrane with an actin antibody. Did you probe the whole membrane with antibodies against ORF2 and ORF3 products? The original blot images uploaded are cropped ones. Add size bars to all images in Fig 5B since they were individually taken.
- There is no UC-rHEV data added to Fig 6. Double check.
Author Response
We sincerely appreciate your thoughtful and constructive feedback, which has been invaluable in further improving the quality of our manuscript. We have carefully considered each of your comments and revised the manuscript accordingly. Our detailed responses to your comments are provided in the attached PDF file.

Reviewer 2 Report
Comments and Suggestions for Authors
The authors have adressed all the points raised and the revised manuscript is now improved.
Author Response
The authors have adressed all the points raised and the revised manuscript is now improved.
Response: Thank you for your favorable evaluation.